# FossilSketch: A novel interactive web interface for teaching university-level micropaleontology

Category: Research

## ABSTRACT

Although the demand for geoscientists is projected to grow and the current population of experts is aging, few students are trained in using micropaleontology. Applications of micropaleontology in solving geologic problems are diverse, and include such areas of research as estimating sea level fluctuations, understanding the causes of past climate upheavals, and finding economically important resources like oil and gas. To aid in teaching micropaleontology in undergraduate classrooms, we developed FossilSketch, a web-based interactive learning tool for the basics of micropaleontology. FossilSketch teaches microfossil identification for Foraminifera and Ostracoda through automatically assessing sketch-based exercises and other practice activities. Results from deploying this system in an undergraduate geology class indicate that FossilSketch benefits both students and instructors. Students find FossilSketch more engaging and less stressful than traditional methods, and instructors have their workload reduced in terms of course preparation.

**Index Terms:** H.5.2 [User Interfaces]: User Interfaces—Graphical user interfaces (GUI); H.5.m [Information Interfaces and Presentation]: Miscellaneous

## 1 INTRODUCTION

Micropaleontology is a critical tool for determining the ages of sedimentary rocks for both industrial and scientific applications [26]. Microfossil species are sensitive to specific environmental parameters and are often used to reconstruct past changes in ocean temperature, coastal sea-level, and seafloor oxygenation [34]. Further, microfossils are used in modern, real-time, environmental monitoring because they respond quickly to environmental change [9]. Additionally, micropaleontology can be used in oil exploration to locate reservoirs [38].

Despite the importance of micropaleontology for geoscience research and industry, most geoscience students are not exposed to this topic. Micropaleontology is rarely taught at the undergraduate level because of the number of contact hours necessary and the amount of instructor feedback required to train students at the necessary level of detail. Thus, although the field of geology has broadened over the last several decades, micropaleontology is being dropped from the curriculum and students' training in the field has correspondingly declined [4, 46]. This trend becomes problematic as experts in micropaleontology are aging and fewer students are being trained in microfossil identification techniques [39].

To enable and enhance the training of undergraduates in the basics of micropaleontology in remote, hybrid, and in-class conditions, we developed FossilSketch. FossilSketch, depicted in Figure 1, is an interactive, intelligent digital tool that introduces students to micropaleontology through educational videos, mini-games, sketch-based identification exercises, and assessments focused on applications of microfossils in the geosciences. Sketch recognition algorithms are used to automatically evaluate sketches and provide feedback to help students internalize various morphological features and identify microfossils from two common microfossil groups, Foraminifera and Ostracoda. FossilSketch reduces the burden on the instructors by providing feedback for these activities and games to help students learn and practice micropaleontology skills. This

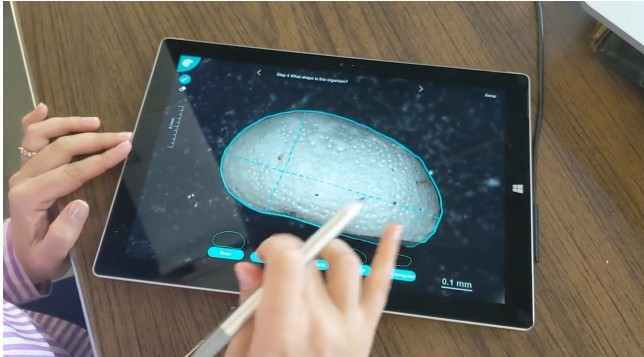

Figure 1: A participant using the FossilSketch educational web app.

paper outlines the design of FossilSketch as well as its impact from being deployed in an undergraduate geology classroom.

## 2 RELATED WORKS

### 2.1 Geoscience Educational Tools

Geosciences have been rapidly adopting online and remote-based educational tools over the last five years, including various online resources, pedagogical practices, and course curricula, including high-resolution digital imaging for mapping and documenting geological outcrops, 3D virtual simulations, digitization of fossil collections, and augmented reality field trip games for smartphones and tablets (e.g., [3, 6–8, 11]).

Successful implementation of software in geoscience education includes sketching software, virtual microscopes, and field experience simulations [8, 18, 31]. For example, CogSketch is a sketching-based application with a series of introductory geoscience worksheets on key geoscience concepts [18] that aids students in solving discipline-specific spatial problems while providing instructors with insights into student thinking and learning.

As for micropaleontology, researchers note a lack of human experts and decline in micropalentology training [10, 25, 32]. That said, most software development has been aimed at automated identification of microfossils, with the most recent approaches focusing on machine learning and using 3D models for planktic and benthic foraminifera identification [10, 25, 32]. Several large microfossil databases were built [13–15, 43]). However, these online resources are designed for an advanced user and are difficult to use for entry-level specialists and students without prior instruction on microfossils.

Until recently, there were no applications supporting active learning in micropaleontology. FossilSketch is the first application that supports active learning in undergraduate micropaleontology [references redacted for anonymization] To summarize, there is clearly a need and growing interest in developing automated AI tools for geoscience education and microfossil identification. To address this need we designed FossilSketch, a novel, universally accessible, and academically rigorous educational tool for undergraduate geoscience education.

## 2.2 Digital Sketch Recognition in the Classroom

Sketching activities in the classroom have pedagogically been linked to enhanced student creativity and learning [33, 35, 40, 50, 52]. Researchers find that sketching benefits learning in a wide range of disciplines, from human anatomy and biology to engineering, geography, and math [5, 18, 19, 37, 44]. Studies have confirmed that information retention and learning outcomes are significantly improved when engaging in drawing and writing activities vs. using a keyboard as the primary input modality [33]. To that point, sketch-based learning tools have been linked to a higher retention of information and improved skill compared to students who do not learn with sketch-based activities [21, 53].

Early gesture recognition systems developed by Rubine [45] have led to improved recognition systems including template-matching algorithms from the "Dollar" family of recognizers [1, 2, 48, 49, 54] that produced lightweight recognition systems easily added to existing software. The "Dollar" recognizers perform classification tasks by using different methods of calculating distance from user-generated input compared against several samples of trained data. Despite these recognizers being used for classification techniques rather than grading sketch accuracy, we use this work as a basis for our recognition system due to synergy in design. Both feature-based classification techniques and template matching techniques were later expanded into more robust systems for scaffolded recognition via systems like *PaleoSketch* [41] and *LADDER* [22], the second of which is notable for its integration of domain-specific shapes to better describe relationships between sketch properties to assist in recognition. More recent works like *nuSketch* [17] and *COGSketch* [16] integrate sketch recognition algorithms into educational tools to assist with the learning experience to measurable success.

*Mechanix* [36, 47], *Newton's Pen* [30] and *Newton's Pen II* [29], *Physics Book* [12], and *SketchTivity* [23, 51] are systems specifically written to leverage the educational advantage of drawing and sketching into the core interactions of their tools. Indeed, these systems serve as the primary conceptual basis from which FossilSketch is designed. We aimed at adapting the educational techniques presented by these tools to the domain of micropaleontology in the classroom. This led to a variety of changes and design considerations taken in the teaching approach outlined in the next section.

## 3 DESIGN

### 3.1 Design Considerations

FossilSketch is a web-based educational tool for teaching students techniques for identifying microfossils. FossilSketch focuses on Foraminifera and Ostracoda due to their utility and accessibility in undergraduate lab settings. Foraminifera and Ostracoda are two of the most commonly used groups of microfossils in industrial, environmental, and scientific applications. The morphology of species in both groups is closely related to the environments in which they live [20, 27, 42] and these two groups are often used in species-specific geochemical studies [24]. However, accurate species identification is required for using this micropaleontological tool effectively. For additional context, Foraminifera are amoeboid protists with shells made of calcium carbonate or agglutinated sediment grains and are often abundant in marine environments [4], and Ostracoda are micro-crustaceans with a bivalved calcareous carapace that are found in all aquatic environments from fresh water lakes to to the deep-sea [4]. These are also some of the larger microfossils, which allows students to view them with standard stereoscopes.

To use industry standard packages and tools, the website is built using the Next.js framework and a MySQL database. Educational materials for FossilSketch were developed to supplement various geoscience courses in the College of Arts & Sciences at a large R1 university. Traditionally, undergraduate students learn about micropaleontology through lectures, diagrams, specimens viewed through a stereoscope, and hand-sized models in upper-level courses as part of paleontology courses. FossilSketch educational materials include the following: 1) educational videos; 2) instructional mini-games; 3) microfossil identification exercises; and 4) microfossil assemblage reconstruction exercises. All four types of activities consist of content specifically created for FossilSketch, based on real-life scientific study cases, and tailored to support the educational exercises in traditional and FossilSketch-based courses.

Exercises were developed based on the courses' learning objectives, the microfossil collections available, and the expertise of [co-author names redacted for review]. FossilSketch can be used in courses of different levels, from lower-level non-geology majors to upper-level courses for geology majors, and thus, the difficulty and number of activities included in a class vary depending on the teaching goals and the activities assigned to students. Modules, and microfossils can be added, or removed depending on the class or activity in which FossilSketch is deployed. The self-contained nature of the exercises and the flexibility of the landing page interface offers the versatility of rearranging the website experience depending on the course learning objectives.

### 3.2 Educational Videos

Educational videos were created specifically for FossilSketch and were created to provide introductory information to help contextualize concepts covered in the rest of FossilSketch's activity types [links redacted for anonymization] When users click on these modules, an overlay with an embedded YouTube link is displayed. Students are free to change playback with the standard embedded YouTube video controls, and captions, and the overlay can be dismissed at any time by clicking outside of the video area. No progress data is recorded for this type of activity.

FossilSketch is intended to augment instructor lectures, meaning the videos are not intended to serve as a replacement for lecture material as is usually the case with typical instructional videos in an online learning interface. The FossilSketch system uses instructional videos to provide necessary information for students to engage with the rest of the modules if the students have not yet received instructor lectures, while at the same time emphasising concepts most directly relevant to the activities if they have attended in-depth lectures in the classroom.

### 3.3 Instructional Mini-Games

FossilSketch integrates various kinds of interactive instructional tools. In order to improve student comprehension of microfossil identification, we broke identification tasks into mini-games that students could repeat to develop mastery. Each mini-game consists of one or more types of interactions intended to highlight the visual-morphology aspect of learning about microfossil identification. We currently have three matching games and one orientation game.

#### 3.3.1 Matching Games

Matching games require the participants to match morphological features, such as the outline shape for Ostracoda, or morphotype and type of chamber arrangement for Foraminifera. At the beginning of the game, the students are presented with a reference image that lists each morphotype along with a sketched example, and students are able to return to this reference image again, when needed, by clicking on the zoomed-out image on the bottom right corner of the screen. When the game starts, the screen displays a small number of draggable "discs" or rectangular "cards" with actual microfossil photomicrographs that the user can move into slots with sketched categories for each feature used in this game. At the moment, three different mini-games are created with this kind

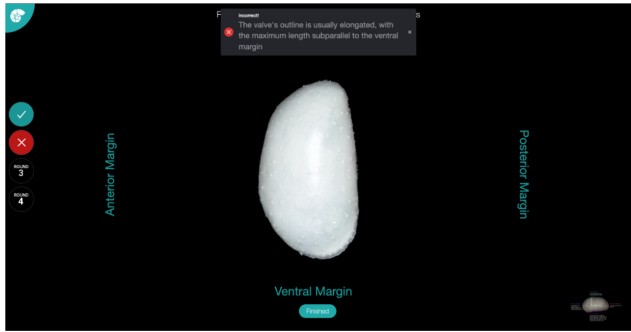
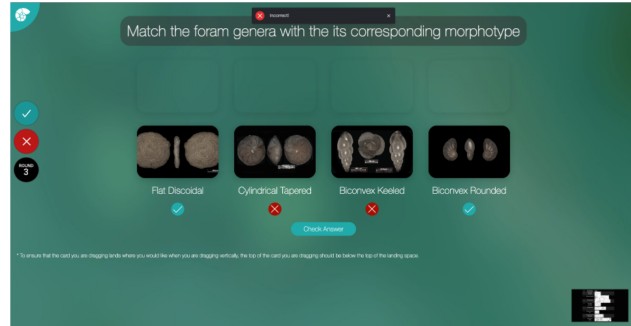

(a) An example of the Ostracod Orientation Game. In this example, the student got the answer incorrect and receives feedback to help guide them to the correct orientation.

(b) An example of the morphotype matching game. In this example, the student can see which two morphotypes they got incorrect.

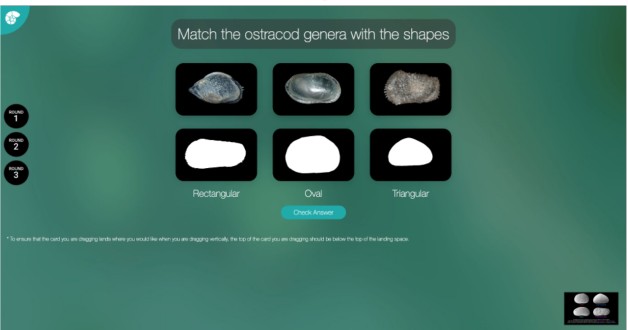
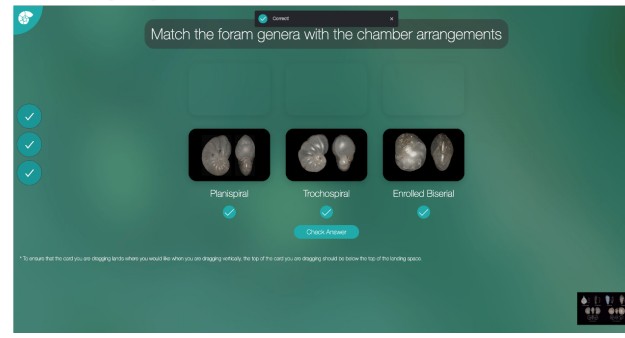

(c) An example of the Ostracod Outline game. This image shows the game before the student has started playing.

(d) An example of the chamber arrangement matching game. In this example, the student has gotten everything correct and will momentarily get feedback indicating how well they did on the exercise overall.

Figure 2: FossilSketch mini-games

of interaction: Ostracoda lateral outline identification, Foraminifera chamber arrangement, and Foraminifera morphotype identification.

All matching games include three rounds, with each level contributing to a final star score. The Foraminifera chamber arrangement mini-game randomly pulls images of Foraminifera from the database for matching to the corresponding chamber arrangement types, with each round of the game having four cards to match. In the morphotype mini-game, the number of draggable items and slots in later rounds increases from 4 in the first round, to 8 in the third round to increase difficulty. If the answer is incorrect, FossilSketch provides a hint by showing a hint or indicating which of the cards were matched incorrectly, and a user can try again to submit a correct answer. Students receive a star rating from one to three based on how many rounds they got correct on their first attempt.

### 3.3.2 Orientation Game

The orientation game integrates a rotation interaction to help students gain an understanding of how to correctly orient the ostracod valve for identification. An ostracod valve has four sides: dorsal, ventral, posterior, and anterior margins/sides. This game starts with a general description of each of these margins to help students gain an intuition of how to identify each side of an ostracod. The user is tasked with rotating an ostracod to its position with the dorsal side up and all of its sides correctly labeled. To simplify the interaction, students rotate in one direction 90 degrees at a time by clicking or tapping once on the ostracod that is displayed in the center of the screen. When the student believes that the ostracod is oriented correctly, they submit their answer by selecting the "Finished" button on the center bottom of the screen.

As in the matching games, the orientation games are divided into three rounds. In this case, each round consists of one ostracod valve that needs to be rotated into the correct orientation. Answers are

marked "correct" if they are rotated correctly the first time. If the submitted answer was incorrect, FossilSketch provides a hint on how to orient the valve correctly. Students are encouraged to use the knowledge gained from the hint by correcting their wrong answers. The star rating is based on the first submitted attempt for each round.

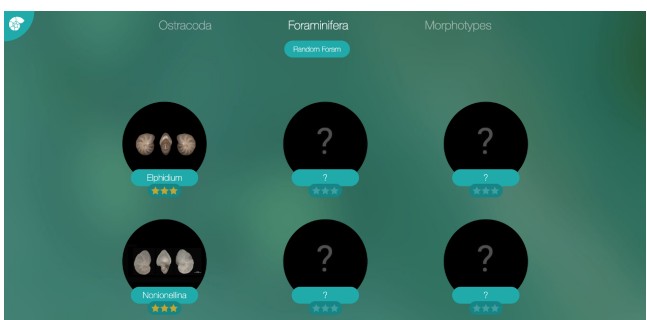

Figure 3: Menu of the morphotype ID exercises. Students pick from any of the unidentified morphotypes marked with a "?", and afterwards are shown their performance on a 3-star rating system.

### 3.4 Identification Exercises

In micropaleontology, microfossils are picked from sediment samples and the obtained variety of different species represents an assemblage characteristic of the sample and may point to the environmental setting or geologic age of the sample. A micropaleontologist would identify the species of microfossils in

this assemblage based on their morphology, or their characteristic features. Primarily, FossilSketch offers a scaffolded learning experience to guide students through the steps needed to identify microfossils and their morphological characteristics.

Students are first presented with a menu depicted in Figure 3, where they can select to identify a specimen of Ostracoda or Foraminifera to genus level or a morphotype of Foraminifera. The Foraminifera identification steps to genus level can be seen in Figure 4 and are the following: 1) sketch the outline of the foraminifer image on the left; 2) sketch the outline of the last chamber on the image on the left; 3) select the type of shell this foraminifer has; 4) choose the overall shape of the organism from a menu; 5) choose the shape of the chambers; and 6) their number; 7) choose the type of chamber arrangement from the menu; 8) select the aperture location from the menu; 9) and select the aperture shape from a menu; 10) identify a genus based on the selected features. The Ostracoda genera identification exercise steps are shown in Figure 5 and include: 1) sketch the maximum length of the valve; 2) sketch the maximum height of the valve; 3) identify right vs left valve; 4) sketch the outline of the ostracod valve; 5) choose the type of outline from the menu; 6) measure approximate size of the valve and choose the size range from the menu; 7) choose the types of ornamentation, select any additional features when present; 8) and identify an ostracod genus based on the selected features.

Within each exercise the types of interactions are described below:

### 3.4.1 Sketching Interactions

Sketching (steps 1-2 for Foraminifera, and steps 1-2 and 4 for Ostracoda) helps students retain and understand the various shapes and outlines they observe in different microfossils. It is the primary method of interaction after which the project is named. Sketching interactions integrate functionality from a library called paper.js to deliver flexible drawing interactions. Although the system is intended to be used with styli and touch to most naturally resemble a sketching activity, it is also possible to draw with a mouse or trackpad. Drawing interactions are usually integrated as the first steps of both kinds of identification exercises, as the overall shape of the sample is critical in identifying the microfossil.

The FossilSketch system checks for correctness using a template matching algorithm, outlined in Algorithm 2. The template recognizer coded specifically for FossilSketch uses the Hausdorff distance metric to determine the accuracy to the key for each microfossil. Before recognition, both the template and the input sketch are resampled to a lower sampling rate with roughly equidistant points as outlined in Algorithm 1. The formula followed for calculating the interspace distance is given in Eq. 1 where $c = 256$ is a constant empirically derived to adjust the distance between the points for optimal calculation of the distance metric. The algorithm then iterates through each point in the input sketch, comparing it with the corresponding point for the template sketch and calculating the Euclidean distance between the two. Total distance is calculated across all the compared points and the cumulative sum is the overall "distance" between a template and the student input (see Figure 6). If the average deviation of the points is greater than the pixel with of the canvas divided by a constant, the algorithm concludes that the input sketch is too different from the template sketch. This constant was empirically determined after internal testing to match the desired student experience; students are meant to provide a relatively accurate, but not perfect, recreation of the template.

The template sketches are provided by [co-author names redacted for review] and coded directly into each foraminifer or ostracod image. Every foraminifer has a database entry containing template sketch data and the outline for its left view (see Figure 3 step 1), and its last chamber (Figure 3 step 2). For every ostracod in a database, there is a template sketch data for the outline, maximum length, and maximum height.

$$S = \frac{\sqrt{(x_m - x_n)^2 + (y_m - y_n)^2}}{c}, \; c = 256 \qquad (1)$$

---

**Algorithm 1** Resampling Technique

---

**Require:** Point list *path*, distance $S$
**Ensure:** Re-sampled point list *out*
  $D \leftarrow 0$
  **for** $i$ in *path* **do**
    $BetweenDist \leftarrow \sqrt{(x_{i+1} - x_i)^2 + (y_{i+1} - y_i)^2}$
    $D \leftarrow D + BetweenDist$
    **if** $D > S$ **then**
      $D \leftarrow BetweenDist$
      $out \leftarrow$ new point $(x_i, y_i)$
    **end if**
  **end for**

---

---

**Algorithm 2** Compare Sketches

---

**Require:** Student $Spath$, template $Tpath$
**Ensure:** Boolean *result*
  $totalDeviation \leftarrow 0$
  **for** $i$ in $Spath$ **do**
    $closestDistance \leftarrow INF$
    $longestIndex \leftarrow 0$
    **for** $j$ in $Tpath$ **do**
      $tempDist \leftarrow$ distance between $Spath_i$ and $Tpath_j$
      **if** $tempDist < closestDistance$ **then**
        $closestDist \leftarrow tempDist$
        $closestIndex \leftarrow j$
      **end if**
    **end for**
  **end for**
  $avgDeviation \leftarrow \frac{totalDeviation}{spathlength}$
  $cwidth \leftarrow$ pixel width of canvas
  **if** $avgDeviation > \frac{cwidth}{70}$ **then**
    $result \leftarrow$ True
  **else**
    $result \leftarrow$ False
  **end if**

---

### 3.4.2 Identification of Features from a menu

Identification of features (steps 3-5 for Foraminifera, and steps 3, 5-6 for Ostracoda) is presented to students as a horizontal multiple-choice menu along the bottom of the screen. During each of these steps, the student is asked to identify one of several characteristic features of the microfossils. For instance, the student might be asked *"what is the overall shape of the organism?"* and the possible answers might be *"vase-like", "convex", "low-conical", "spherical"* and *"arch"* among others. With each option, a sample sketched outline of each shape is shown, but it is important to note these are sketched examples and not photorealistic depictions of the choices. The student is tasked with remembering the particular physical properties of each characteristic feature, as well as matching the pictures with the closest choice from the menu. Of these, one is the correct answer. In this part of the exercise, the student does not receive immediate feedback to their submitted selections, and all of these answers are summarized for the student to use in order to make the final identification from the database of genera for Foraminifera and Ostracoda.

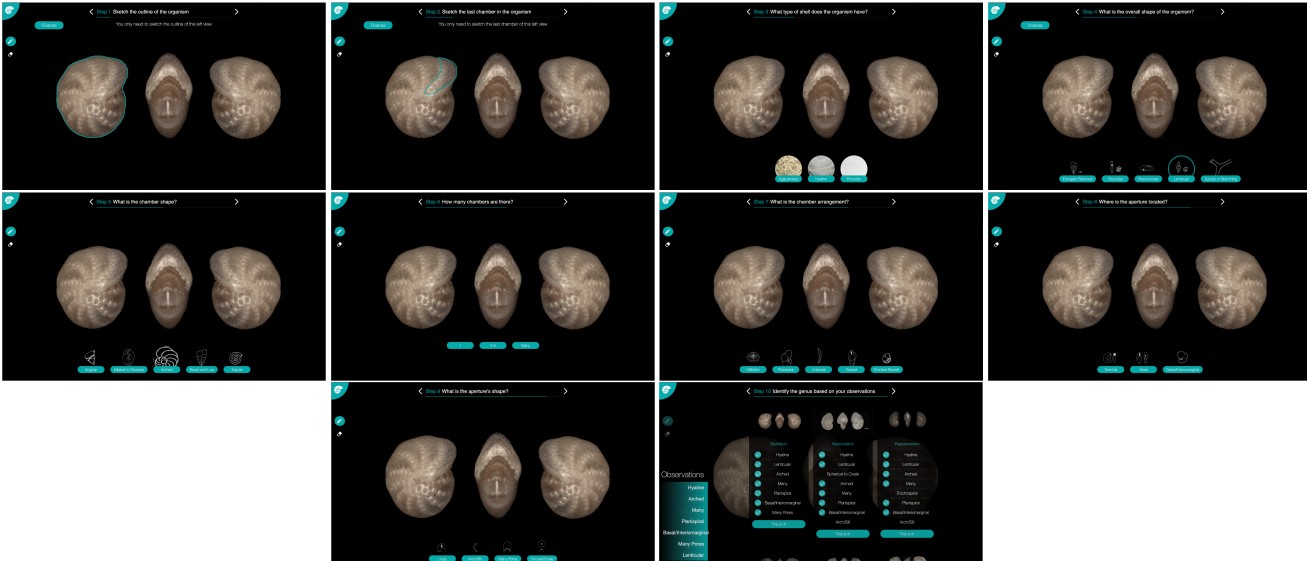

Figure 4: Foraminifera Identification Steps (from left to right, top to bottom): 1) Sketch the Outline, 2) Sketch the Last Chamber, 3) Select the Shell Type, 4) Select the Overall Shape, 5) Select the Chamber Shape, 6) Select the Number of Chambers, 7) Select the Chamber Arrangement, 8) Select the Aperture Location, 9) Select the Aperture Shape, 10) Identify the Genus

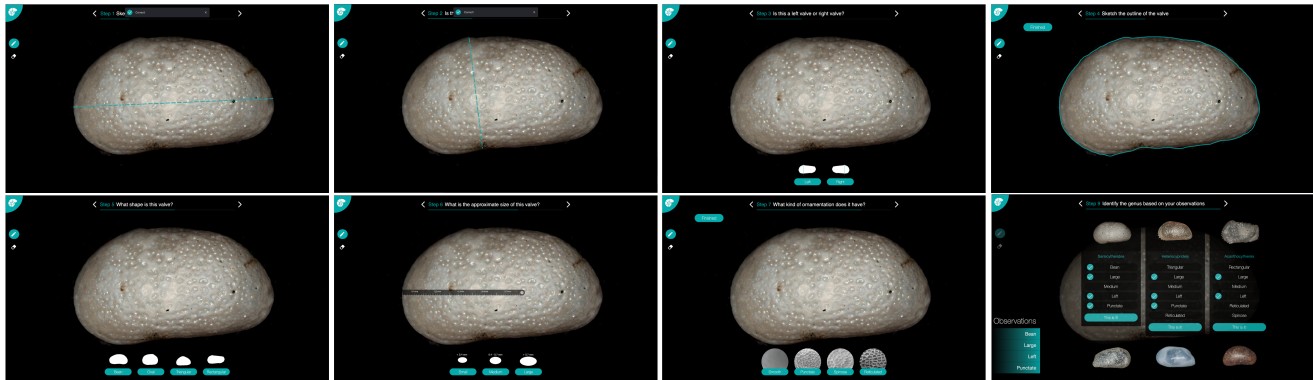

Figure 5: Ostracod Identification Steps (from left to right, top to bottom): 1: Sketch the Max Length, 2) Sketch the Max Height, 3) Identify right vs left valve, 4) Sketch the Outline, 5) Select the Valve Shape, 6) Select the Approximate Size, 7) Select the Ornamentation, 8) Identify the Genus

### 3.4.3 Pointing Interaction

Pointing interactions (step 5 for Foraminifera morphotype ID) are a simplified form of "sketching interactions" that require students to click once in a general area of interest, and FossilSketch checks if the identified location is correct. Specifically, this interaction is used to identify the general location of the aperture of a given foraminifer. The student is asked to click once in the region where they believe the aperture is. Each foraminifer in the FossilSketch database contains data on a rectangular region that points to the general area of its aperture. When the student clicks "Submit" after identifying the aperture area, FossilSketch checks to see if the location of the click is within the predefined rectangular area. If it is, the answer is marked as correct. The location of the aperture is only used for identifying a foraminifer's morphotype.

### 3.4.4 Summary Screen

The summary screen (step 10 for Foraminifera, and Ostracoda) is the last step for each identification exercise, asking the student to draw from their observations and make the final selection of the genus or morphotype for Foraminifera or Ostracoda. Each morphotype or genus has a list of characteristic features, and, based on student answers, each feature correctly marked during the identification steps would have a green check mark. The list of morphotypes or genera on the summary screen is ranked by the highest number of matching properties with student answers. If the student's answers are correct, the choice is easy since it has the most check marks and is the first item listed. Additionally, a picture of each morphotype of genus is included, letting students double-check to see if their best-ranked choice is the most accurate. This system allows students to develop self-assessment skills to see if their choices match up with any given morphotype or genus. At any time students are able to revisit any of the previous steps, so this final choice would be a good motivation to do so if they notice their prior choices did not yield a definitive conclusion. It also allows students to see different properties that might be common between some morphotypes or genera, but each foraminifer and ostracod specimen will have only one correct final answer.

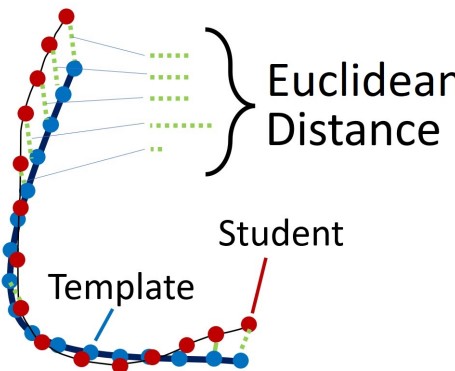

Figure 6: To evaluate answers, FossilSketch resamples and overlays both the student input and instructor-provided sketch, and a total distance metric is calculated by summing the Euclidean distance between sampled points.

### 3.5 Assemblage exercise

One of the goals of this interface is to demonstrate to students the various applications of microfossils in geosciences. Once the students gain mastery of microfossil identification through practicing mini-games and microfossil identification, they proceed to the final type of exercise and assessment where they can apply their knowledge to reconstruct environments from an assemblage of different microfossils. In this exercise, the students view microfossil assemblages with approximately 20 foraminifer or ostracod individuals and identify the foraminiferal morphotypes or Ostracoda genera present. These assemblages imitate an actual microfossil "slide", as seen under a microscope that contains an assemblage of Foraminifera or Ostracoda. Students are asked to identify how many of each foraminiferal morphotype or ostracod genus specimens are present in the slide. Before students start working on the exercise, they can view a screen with a summary of the information on foraminiferal morphotypes or ostracod genera and how they can be used to interpret environmental properties, such as the oxygenation or salinity of the water. This exercise includes 3 rounds and a summary. The student then needs to identify the different genera or morphotypes and select from the menu on the right side of the screen the number of each morphotype. It is intended that students will draw on their knowledge from the previous exercises to quickly identify the morphotypes or genera they see in these assemblages. For the ostracod assemblages, the menu to select from includes both the genera that are and genera that are not present in the assemblage. For the foraminiferal morphotypes, the assemblage includes two morphotypes to select from and "Other" category. To answer correctly, the student must provide a correct number for all categories, i.e., for both of the morphotypes or genera and the "Other" category, in an assemblage.

Both assemblage exercises conclude with a summary page where the student is asked to make an overall conclusion about the environment-based morphotypes and genera present in the assemblages. For instance, the Foraminifera morphotype assemblage exercise uses assemblages to determine bottom water oxygenation. It has been shown that in environments where cylindrical- and flat-tapered morphotypes are found in abundance, the environments usually have low oxygenation [28]. The students are asked to rank each assemblage by relative oxygenation level. They should be able to do so when they consider the relative abundance of cylindrical-tapered and flat-tapered morphotypes they found in each of the three assemblages. Similarly, for Ostracoda genera, students count the number of individuals of each genera, and determine the bottom

water salinity indicated by each of the assemblages. If a student makes a mistake, FossilSketch provides feedback, by showing which specimens correspond to which genera and morphotypes, so one can correct their response. These exercises show how microfossil research is applied and assess microfossil identification skills learned and honed across all exercises of the FossilSketch system.

## 4 EVALUATION

To test the efficacy of FossilSketch as an effective means of teaching micropaleontology, we conducted a case-control experiment in a "Paleontology and Geobiology" course over two different semesters. As the control, students were taught micropaleontology without the use of FossilSketch in the Spring 2020 semester. As the case, students were taught micropaleontology using FossilSketch in the Spring 2023 semester. We describe the experience of the students from each of these semesters in more detail in the following subsections. As a side note, FossilSketch was used in other semesters in between our control and case groups; however, the tech stack for FossilSketch was completely overhauled prior to its deployment in Spring 2023.

### 4.1 Spring 2020

During the Spring 2020 semester, students participated in three-hour-long laboratory sessions consisting of several specimen-based laboratory activities. Students used 3D physical models and labeled SEM images to study the main morphological features of various Foraminifera and Ostracoda respectively. After completing these activities, students were asked to select a microfossil and provide a labeled sketch of the specimen, identify its morphological features, and ultimately identify its genus. Students were encouraged to work in teams and were allowed to ask the teaching assistant or professor any questions they had.

### 4.2 Spring 2023

During the Spring 2023 semester, students were asked to use FossilSketch along with the in-person specimen-based laboratory activities. Specifically, students were asked to watch the educational videos, play each of the four mini-games, and identify at least three different Ostracoda, and Foraminifera. After completing these activities, students were asked to select a microfossil and provide a labeled sketch of the specimen, identify its morphological features, and ultimately identify its genus.

### 4.3 Participants

A total of 86 students, two TAs, and one instructor (who taught both courses) consented and took part in the study, of which 51 students represent the control group, and 35 represent the test group. The instructor is an author on this paper. Before data collection and using FossilSketch software, participants were given a quick overview of the project and signed consent forms (IRB2019-1218M, expiration date 02/05/2026).

## 5 RESULTS

In both semesters we conducted surveys and focus groups with the students. We also conducted semi-structured interviews with the graduate TAs and the professor to get insights into their experience with FossilSketch. We discuss their feedback in the following subsections.

### 5.1 Student Feedback

After using FossilSketch, students completed an engagement survey where they could give feedback about their experience, what they found effective, and what they found difficult. This survey contained open-ended questions regarding their expectations in the course, how they felt about the micropaleontology activities, and what strategies they employed to complete the coursework. To determine

the impact of FossilSketch on student engagement and enjoyment, we conducted a deeper analysis of the responses to the question "Did you enjoy the micropaleontology activities in this class? Which ones? And what about them were enjoyable?". We coded the answers to this question based on whether the tone was positive, neutral, or negative, as students used this question to either describe things they enjoyed or complain about the things they did not. In the Spring 2020 semester, there were 25 answers to this question with 11 being positive, 8 being neutral, and 6 being negative. In the Spring 2023 semester, there were 22 answers to this question with 18 being positive, 2 being neutral, and 2 being negative. Conducting a $\chi$-squared analysis showed that the answers are statistically significantly different with $p < 0.05$. As the two main changes were the increase in positive responses and the decrease in neutral responses, we hypothesize that FossilSketch won over students who had less initial buy-in for learning microfossils. There were students who were notably passionate and critical about learning the material in both groups, which can be expected in any course. Many students were being exposed to microfossils for the first time, so they had little expectation of the utility of learning this tool. For the traditional methods, some of these students left the unit lukewarm, saying that they did not hate the material but also did not enjoy it. By contrast, most students who used FossilSketch answered specific features they liked the most, and several also described the traditional lab activities that FossilSketch augments. In short, FossilSketch was more effective in engaging students to learn about micropaleontology when compared to using traditional methods alone.

Students also demonstrated engagement with FossilSketch through their usage patterns. Several students completed the genus identification exercises for additional practice, with a small number of students completing the exercises six times more than required to complete the lab assignments. The majority of students also indicated that the genera identification exercise was their favorite activity in FossilSketch, because they enjoyed sketching and following step-by-step instructions. Ostracoda exercises were notably more popular with half of the students completing extra identifications (students were required to complete 3 Foraminifera and 3 Ostracoda genera identifications), likely because they are easier to complete due to having fewer steps. A similar pattern arises when looking at the mini-game playing statistics. Half of the students would play matching games additional times. The most difficult exercise, the assemblage exercise, was only occasionally played additional times, but this result is expected due to its difficulty.

### 5.2 Teaching Assistants' Feedback

We conducted a semi-structured interview with the Teaching Assistants (TA) from both the Spring 2020 and Spring 2023 courses to understand how FossilSketch impacted their experience.

#### 5.2.1 Spring 2020

Overall the TA was quite negative about the experience of teaching microfossils. The TA has to learn the material beforehand from the instructor in order to properly proctor the lab session. The instructor explains what the answers are to the lab questions and what to look for in the specimens to identify them so that the TA can answer questions during the lab. This preparation is necessary as recognizing the different microfossils is challenging without experience. To that point, the TA noted that the students found the topic difficult to grasp:

> *"Challenging, some students were very confused. Some students were okay, but some found it really hard to understand, as compared to other groups [macrofossils]. Microfossils were definitely more difficult for them."*

She went on to note that, given the difficulty students have in learning about microfossils, more time needs to be spent on

teaching the subject. Learning the different species requires gaining familiarity with the unique features and attributes, which involves getting exposure to samples and practicing identifying them. Furthermore, fully understanding and committing these concepts to memory can require significant creativity.

> *"You have to be creative talking to students, like coming up with some non-traditional ways to remember morphology features, like: Uvigerina looks like a banana bunch, just imagine that. I used a lot of imagination when I was trying to grasp that."*

#### 5.2.2 Spring 2023

Overall the TA was positive about her experience with FossilSketch being used as part of the lab assignments. She felt that students benefited from its use and that it sped up the process of learning about microfossils.

> *"I did have one student tell me that this was the least confusing lab out of all of them. I thought that was pretty amazing. So I think it is very good for a kind of helping me kind of an abstract idea into actually something tangible for people to understand."*

Regarding her experience as a TA, she noted that using FossilSketch lightened her workload, as students asked fewer questions overall and she could rely on FossilSketch as a tool for answering some of the questions that did arise. FossilSketch provided a database to look up visual aids as well as a medium to walk through the identification process.

> *"I think it made my work easier. People kind of just went off on their own, and they kind of worked through it on their own. [...] All I did was I put up the key that was in the corner of one of the mini-games. I just went up there and said like, "Look at this." So then they could actually figure it out from there. So that was really helpful."*

She mentioned that the students did come to her with some bugs and issues with the software, but these did not detract meaningfully from the student's overall experience using FossilSketch. She noted that as a graduate student herself she could see herself using FossilSketch as a reference, and she felt that leaning into this idea of FossilSketch as a reference could make the website more broadly useful. For instance, she suggested adding a glossary of terms with images and examples for students to conveniently reference the basics.

### 5.3 Instructor Feedback

We conducted a semi-structured interview with the professor who taught both the Spring 2020 and Spring 2023 courses to understand how FossilSketch impacted her workload and her teaching.

#### 5.3.1 Spring 2020

When asked about the attitudes of students towards microfossils in this class she noted that students were generally quite excited to learn about microfossils; however, there were several sticking points within the class. Students would become quite frustrated when looking at samples through a stereoscope as they were being asked to view and analyze tiny objects that are inherently difficult to see and parse. Furthermore, students complained about having the sketch the microfossils, finding the task quite tedious.

> *"So even with a stereoscope, they're often relatively difficult to see, and so students get very frustrated because we're asking them to notice things and see things,*

*and they're not able to zoom in enough. [...] They also can't turn it over and manipulate it, and that also is a frustration because there's certain anatomical parts of it that you could see best if you could turn it. [...] So I think students find themselves very frustrated and as that level of frustration rises, their ability to learn goes down, right?"*

When asked if there was a difference in experience between ostracods and forams, she expressed that because she is an expert in Foraminifera and is less personally interested in Ostracoda she felt that translated to how well students were learning about the two categories of microfossils. Not only did she teach the two categories differently, going into more detail on forams than ostracods, but she also expressed that she was likely better at making students more comfortable with the former given her own comfort with the subject.

*"I think students feel the same way about each of them [ostracods and foraminifera] because they're tiny, mysterious things, but I'm probably better at making them comfortable with forams just because of my position."*

### 5.3.2 Spring 2023

To integrate FossilSketch into her classroom, the instructor replaced part of the lab assignments and the paper sketching assignments with FossilSketch exercises and games. FossilSketch was generally scheduled to be completed at the beginning of the lab session, although some students would complete the exercises before the lab in preparation. Students were excited to use a computer-based tool.

When asked how students responded to FossilSketch, she noted that students appreciated a number of its aspects. She noted that students enjoyed being able to go back and do something over again, reviewing microfossils as many times as they needed before dealing with the physical specimens. They also appreciated being able to do their assignments and review the material from anywhere and at their own pace, rather than having to complete the tasks under the pressure and constraints of being in a physical lab and a given time limit. She also explicitly noted that while students would frequently complain about physically sketching microfossils, they did not complain about sketching the fossils using FossilSketch.

She noted that while students still complained about the assignments, their complaints shifted.

*"So I think the difference is where their frustration points are. Before, all of their frustration points were focused on the microscope, and then with the introduction of FossilSketch, their frustration points get focused on the computer. But what I found interesting was their frustration with the microscope declined, so I still had students do things in class looking at the specimens. But I think because they had seen the specimens in another way, they felt more comfortable looking down the microscope."*

She also commented that students' conceptions of how much they could learn changed due to the introduction of FossilSketch.

*"So I've also found that the way that they think about how much they know changed. So like when they were doing the traditional teaching, I think they felt like they knew everything they could know, like the things that they didn't know were just not accessible to them, like the materials weren't good enough. [...] And now they've kind of – they shifted a little bit. To now, they feel like they don't know... I guess bigger things? So like instead of them feeling like they don't really know what a foram*

*is, they feel like they're now focused more on: 'I don't know how to apply them'. [...] So I think they still they still have this feeling that — students always feel like 'I don't know anything, I don't know everything yet, I have to study more'. They always kind of have this feeling. But now that feeling has been transferred to kind of higher level ideas which is actually really useful."*

When asked about the effect of FossilSketch on her own workload, she noted that initially, just like any change to the curriculum, it required effort to develop the materials and figure out how to incorporate them into her specific use case; however, after that initial set-up effort, it was just as easy to incorporate into her classroom as the traditional lab assignments. She did note that FossilSketch did make it easier for her to train the TA, as she could just ask the TA to use FossilSketch. In that sense, she also felt that it lowered TA anxiety, as the TA was not required to know as much of the material as they could (and did) point students with questions to FossilSketch in order to get answers.

Finally, when asked if she would want to utilize both FossilSketch and traditional teaching approaches to teaching microfossils, she mentioned that FossilSketch had distinct advantages in specific scenarios that would lead her to use FossilSketch exclusively, whereas in other scenarios she would want to rely more heavily on traditional approaches. If she were to teach microfossils in an online and/or remote course, she would use FossilSketch primarily. FossilSketch also could be used for students who need accommodations, such as those who cannot look into a stereoscope but can interact with a screen or those who are unable to physically attend in-person labs. She noted that for students that are geology majors, she would want them to physically look at specimens under a stereoscope; however, for students who are non-majors who will likely never look at specimens again, FossilSketch would offer them enough of the material without the frustration of looking through a stereoscope.

## 6 CONCLUSION

FossilSketch is an intelligent tutoring system to support learning micropaleontology in undergraduate geoscience classrooms. The tool teaches students how to recognize Foraminifera and Ostracoda microfossils using sketch-based exercises and mini-games to practice identifying these specimens. We evaluated the effectiveness of FossilSketch in the classroom from the perspective of the instructors and students using qualitative and quantitative analysis. The results show that students respond better to FossilSketch and that the burden on the instructors is reduced, resulting in a better classroom experience for all parties.

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
