# OpenReview forum: "FossilSketch: A novel interactive web interface for teaching university-level micropaleontology"
_graphicsinterface.org/Graphics_Interface/2023/Conference_SD — Submitted to GI 2023 - second deadline_

### Official Review · Reviewer_BH1M · 2023-04-17
**Like the topic but question the generalizability / takeaway messaging**

**Rating:** 6
**Confidence:** 4

**Review:**

This submission is about FossilSketch, a web interface that was designed to teach college/university students about different micropaleontology concepts. The text describes the need for the interface, the four activities that are supported, and the results from the usage of the interface during a course in the 2023 spring semester. The contribution comes from the interface / activities and the outcomes from the course usage. This work does fit the scope of GI.

This was an interesting and well-written paper. It clearly identifies the need for the software, articulates the activities that the software can support, and the findings are from the perspectives of students, a TA, and the course instructor. It is a nice, complete piece of research that presents a software system that I could see many students benefiting from (especially in remote scenarios or if they have accessibility concerns with traditional equipment).

I don’t really have many concerns with this submission other than that I am unsure what the takeaway message is for readers. Yes, this system seems to be useful for instructors and students alike, however, what can the community learn from this design and evaluation? The topic itself is very specific, so it is unclear what aspects of this system could be reused in other systems (if anything). Which techniques worked the best? Should future systems use more sketching-based activities? Only have mini games? Is there any data on the frequency of use or total time used? I guess I am missing details about the specific aspects of this system that developers or designers could use in other educational systems.

I do also wonder what the overarching research question or design guidelines were that guided the development of the software. How was it decided to have videos, minigames, a sketching exercise, etc. Part of the interesting piece about this research is how the different aspects were ideated on, and developed (as the text notes, the system underwent a technical re-development prior to 2023 so it would be useful to provide details about this).

The only other major concern I have is that the submission doesn’t really provide any details about the educational outcomes that were achieved. Were students actually able to learn more / retain more information? While the TA and instructor feedback is useful, and knowing what students thought about the software is interesting, at the end of the day, success comes down to the degree to which students were able to learn and obtain the desired educational outcomes. It would be useful if the text discussed this.

Overall, this submission is an interesting case study of an interactive, web-based application for college / university students. The submission is above the bar enough that it should probably be accepted, however, it may be useful to consider briefly discussing any outcome-based results that were obtained and better identify the specific takeaways from the paper.

---

### Official Review · Reviewer_tcJQ · 2023-04-23
**Interesting system but no real HCI research contribution**

**Rating:** 4
**Confidence:** 4

**Review:**

The paper describes a system for teaching aspects of identification tasks in micropaleontology, using visual presentation of images as well as sketch-based interaction. The paper presents the results of a longitudinal informal evaluation of the system in real classrooms, with most feedback suggesting that the tool is valuable for both teachers and learners. However, even though the system appears to be successful in meeting its objectives, the submission does not adequately identify a research contribution to HCI - the techniques in the system (such as the sketching capabilities) do not appear to be novel, and the system itself does not clearly make advances in areas that would be of value to an HCI research audience.

The paper suffers from a problem common to many systems papers, in that it does not find an area of HCI research in which to claim a contribution. Simply doing a good job of what the system was designed to do is not in and of itself a valuable contribution, and the authors do not clearly identify any area where they have made an advance on the state of the art. For example, the sketch-based interaction does not appear to go beyond the previous work that is cited in the paper - and although the application to the domain of micropaleontology is different from previous efforts, the authors do not clearly indicate how this shift to a new domain led to new insights or techniques.

Systems papers can make contributions in many different ways - identifying interaction requirements, specifying new interaction problems, developing new interaction techniques, providing new engineering artifacts that address issues such as efficiency or latency, or even providing a synthesis-style contribution where no single aspect is ground-breaking but where putting together several already-known techniques represents a substantial advance. However, the authors of the present submission need to do more to identify exactly what their research contribution is, and provide evidence for that contribution.

I note that there may be a contribution here in terms of geology education - but that topic is not within the purview of GI, so is not applicable here.

---

### Official Review · Reviewer_ymDQ · 2023-05-01
**Nice system, but limited research contribution**

**Rating:** 3
**Confidence:** 4

**Review:**

The authors present FossilSketch, an online learning environment to help teach micropaleontology. The system has several types of learning content, including videos, mini-games, identification games, and assemblage exercises. The authors compare a course in Spring 2023 (with FossilSketch) to a course in Spring 2020 which did not use the system. Results suggest that the system was useful.

Unfortunately, I argue against accepting this paper. While the system looks nice and seems to fit a need, I see a limited research contribution. Most of the system follows typical online learning environment content (videos, multiple choice activities), without much additional insight that would be valuable for researchers or practitioners. The most interesting part of the system is the sketching interaction, but this seems to follow typical algorithms and has a limited discussion in the paper. The evaluation is interesting, but not substantial enough to provide research insights to a reader.

In addition, there are several confusing points in the writing of the paper that would need to be improved to help argue for a contribution:
 - In the abstract/intro, the need for micropaleontology being taught is not very strong; most of the argumentation seems to be for increasing geology education, rather than micropaleontology. Why is it critical to support micropaleontology? Is the reduced support for teaching this based on the difficulty for teaching it, or is it simply lower priority?
 - Some of the writing is a little conversational and colloquial ("As for micropaleontology", "That said"), which is a surprising style for a GI paper.
 - In the related work, AI tools are mentioned but the link to FossilSketch is unclear (is AI used here? should it be?)
 - if the videos are not intended to serve as a replacement for lecture material, will this tool actually solve the problem? I believe the argument is that lecturing is not the limiting factor for micropaleontology, but rather lab and feedback hours, but this needs to be crystallized (linking motivation to design decisions).
 - Sketching "helps students retain and understand" -> needs citation or more evidence (I believe it, but it's an unsupported claim)
 - the algorithm doesn't seem to match Fig. 6. In Fig. 6, there seems to be a one-to-one matching for points in the student sketch vs the template, whereas the matching algorithm seems to look at the closest point (which I think is more appropriate). This needs to be more precise, and again linked to motivation/design rationale.

I like this work as an educational tool, but to publish this at a research venue like GI, I believe that it needs to provide useful information that would be novel, surprising, or richer than typical education techniques; to do this, there needs to be novel design or study findings, and the design descriptions need to be clearer (explaining design decisions, why they were made, whether there was insight from piloting, etc.). As it stands, most of the system seems to be typical of learning systems, and the study provides limited findings beyond general support for the tool.